# Design of AD Converters in 0.35 µm SiGe BiCMOS Technology for Ultra-Wideband M-Sequence Radar Sensors

**DOI:** 10.3390/s24092838

**Published:** 2024-04-29

**Authors:** Miroslav Sokol, Pavol Galajda, Jan Saliga, Patrik Jurik

**Affiliations:** Department of Electronics and Multimedia Telecommunications, Technical University of Košice, 042 00 Kosice, Slovakia; miroslav.sokol@tuke.sk (M.S.); pavol.galajda@tuke.sk (P.G.); jan.saliga@tuke.sk (J.S.)

**Keywords:** ultra-wideband, UWB, analog-to-digital converter, ADC, ASIC, SoC, SiP, radar

## Abstract

The article presents the analysis, design, and low-cost implementation of application-specific AD converters for M-sequence-based UWB applications to minimize and integrate the whole UWB sensor system. Therefore, the main goal of this article is to integrate the AD converter’s own design with the UWB analog part into the system-in-package (SiP) or directly into the system-on-a-chip (SoC), which cannot be implemented with commercial AD converters, or which would be disproportionately expensive. Based on the current and used UWB sensor system requirements, to achieve the maximum possible bandwidth in the proposed semiconductor technology, a parallel converter structure is designed and presented in this article. Moreover, 5-bit and 4-bit parallel flash AD converters were initially designed as part of the research and design of UWB M-sequence radar systems for specific applications, and are briefly introduced in this article. The requirements of the newly proposed specific UWB M-sequence systems were established based on the knowledge gained from these initial designs. After thorough testing and evaluation of the concept of the early proposed AD converters for these specific UWB M-sequence systems, the design of a new AD converter was initiated. After confirming sufficient characteristics based on the requirements of UWB M-sequence systems for specific applications, a 7-bit AD converter in low-cost 0.35 µm SiGe BiCMOS technology from AMS was designed, fabricated, and presented in this article. The proposed 7-bit AD converter achieves the following parameters: ENOB = 6.4 bits, SINAD = 38 dB, SFDR = 42 dBc, INL = ±2-bit LSB, and DNL = ±1.5 LSB. The maximum sampling rate reaches 1.4 Gs/s, the power consumption at 20 Ms/s is 1050 mW, and at 1.4 Gs/s is 1290 mW, with a power supply of −3.3 V.

## 1. Introduction

In the modern world of digital technology and digital signal processing, the conversion of analog data into a digital format has become a fundamental part of electronic systems. As part of ultra-wideband (UWB) sensor systems, the analog–digital (AD) converter represents an essential interface between the analog and digital parts of the system. The AD converter is a prime example of mixed, analog, and digital electronic structures. Analog structures represent input amplifiers, comparators, and integrators, which are mostly implemented in fast bipolar or CMOS technology and are based on differential structures. The digital structures are represented by various combinational logic gates, sequential circuits such as the D flip-flops, and clock signal distribution and division circuits. Most AD converter applications can be classified into five general segments. The first segment is data acquisition [1,2], the second segment includes audio and voice applications [3,4], and the third segment includes industrial measurements [5,6,7]. Subsequently, there is control or feedback digital control, and the last segment involves very fast applications such as high-frequency devices [8,9], video processing [10], etc. [11,12].

Several AD converter topologies vary regarding conversion, resolution, sampling rate, structure size, price, and the main purpose of the mentioned applications [12,13,14,15,16,17,18,19,20]. A brief graphical overview of the topologies, depending on the sampling rate, resolution, and purpose is shown in Figure 1 [11].

More detailed information on the basic parameters of AD converters such as gain and gain error, integral nonlinearity (INL), differential nonlinearity (DNL), total harmonic distortion (THD), signal-to-noise and distortion (SINAD), effective number of bits (ENOB), and spurious-free dynamic range (SFDR) can be found in [11,21].

## 2. Current State of the AD Converter Application in the UWB M-Sequence Sensor System

Within the basic UWB M-sequence sensor system, the analog part has been implemented in two semiconductor technologies; see [22,23,24]. These analog parts contain track and hold sampling circuits, the output of which drives the AD converter. There are two main outputs from the analog part of the sensor system, namely, the sub-sampled signal itself and the clock signal for the AD converter. The sub-sampled and clock signals have a frequency bandwidth of approximately 500 MHz.

The presented UWB sensor systems use a commercially available AD converter. Within the range of signal frequencies used in UWB M-sequence systems, commercially available converters can be used without any significant issues. This converter is implemented in the system as a separate monolithic encapsulated component on a single PCB together with the analog part of the UWB circuit chip; see Figure 2. Because the converter has different supply voltages and different signal ranges, it is DC-decoupled from the analog part.

A commercial converter, such as the one shown in Figure 2, increases the PCB design requirements, degrading the parameters and increasing the system size. Therefore, our main aim was to shrink and integrate the UWB sensor system as much as possible while achieving comparable or better parameters compared to the current state-of-the-art systems. One solution is to integrate the converter with the analog part into a system-in-package (SiP) or directly on a system-on-a-chip (SoC). Within commercially available ADCs, it is not easy to obtain an unpackaged, naked die. For this reason, there was a need for customer-specified, custom-designed AD converters.

The reasons for developing a customer-specified AD converter design can be specified as follows:Naked die chips of commercial converters are unavailable to package with the existing UWB system, creating a SiP.Although it is possible to purchase the mask design of an AD converter from third parties, implemented in some technology, this can either be a new design customized specifically for the customer or a clone of an existing converter core with customized I/O circuits for the existing UWB system to create an SoC. However, the purchase of the license and design of such a converter typically costs hundreds of thousands of Euros.The unit cost of a custom converter in semiconductor technology.For the already existing analog part of the UWB system, it is possible to customize the input and output circuit levels in the case of a customer-specified converter.Customer-specified adjustments of converter timing.Easier integration of SiP and SoC.Custom compatible chip pinout.Implementation and possible design adaptation to other technologies.

On the other hand, the disadvantages of custom design are specified as follows:Difficulty and complexity of the customer-specified design, where both analog and digital circuits need to be considered.Emphasis on the design of I/O matching to the analog part of the UWB system, while possibly overlooking/neglecting the core of the converter itself and the resulting inferior AD converter parameters, such as resolution and dynamic ranges.Time and personnel requirements.

Based on the requirements of the UWB sensor system, a parallel converter structure was chosen to achieve the maximum possible bandwidth in the proposed semiconductor technology. A 5-bit parallel flash AD converter was initially designed as part of the research and design of radar systems. The proposed analog input frequency bandwidth of the converter was up to 300 MHz and the maximum sampling frequency was 105 MHz. The effective number of bits was only 2.6 bits. The proposed 5-bit converter is shown in Figure 3.

In the tests and measurements of the AD converter, several design weaknesses were found, such as the on-chip placement of electronics devices, power, and clock signal distribution, as well as the input comparators themselves [16,26].

## 3. Development and Progress of the AD Converter Designs in 0.35 µm SiGe BiCMOS Technology

Parallel to the development of the AD converter for UWB systems, analog UWB systems emitting M-sequences were also developed. The individual systems were implemented in two semiconductor technologies: 0.35 μm SiGe BiCMOS technology from AMS [27] and 0.25 μm SiGe BiCMOS technology from IHP [28]. These analog systems include T&H sampling circuits. Almost all broadband and clock signals are differentially designed for better system robustness. One of the newly designed wideband UWB sensors [27] was explicitly designed using the same semiconductor technology as past designs of AD converters, which increased the motivation and interest in designing a new AD converter. This compatibility also raised the possibility of integrating both the analog part and the AD converter into a single chip, using the relatively inexpensive 0.35 μm SiGe BiCMOS semiconductor technology. Such integration would not only reduce the cost of the wideband sensor system but also simplify the application and commercial use of these systems. Additionally, a significant requirement for applications in UWB M-sequence systems is the absence of sampling T&H circuits in the AD converter, as the analog part of the system already contains these circuits.

Based on the considered applications, we can summarize the requirements of the AD converter design as follows:Maximum number of bits per chip area, 2 × 2 mm.Input voltage range min. 1 V_pp_.Lowest possible power consumption in parallel structure.Power supply compatible with the analog part of the UWB sensor system, −3.3 V.Input circuit frequency bandwidth min. 400 MHz.Differential inputs.Compatible with the analog sampling system part (without sampling circuits).Sampling frequency up to 100 MHz.Outputs compatible with negative LVCMOS standard [29].

Building on the knowledge from the initial design of a 5-bit AD converter and the requirements of the proposed UWB M-sequence system, the design process for the individual circuit structures of the new AD converter started. First, a test 4-bit structure was designed to evaluate the concept of the proposed AD converter structure for specific UWB M-sequence systems. After confirming its sufficient characteristics, a 7-bit AD converter was designed for possible applications based on the requirements of UWB M-sequence systems. A flow chart showing the history and design workflow of the aforementioned AD converters is shown in Figure 4.

The concept of a 4-bit experimental AD converter cell was first developed as a preparation for the design of a multi-bit converter. A block diagram of the 4-bit structure is shown in Figure 5. The structure consists of a main block where the comparators, the reference resistor network, the D-Flip-Flop circuits, and the clock signal distribution circuits are located. The data output of the test converter is output directly in the thermometer code for direct control and measurement of the AD converter core without the influence of the other data processing circuits [30,31,32]. The converter concept also includes an experimental internal “band-gap” current reference to create a reference voltage on the resistor network [33].

The structure of the 4-bit parallel architecture is formed by an array of fifteen comparators. The comparators compare the reference voltage created by a resistor network with the RF signal at the second input. The network consists of 14 resistors connected in series. Each node of the resistor network includes a blocking capacitor to reduce the crosstalk of the high-frequency signal, which can affect the voltage stability at each level of the resistor network. This crosstalk is created by the parasitic capacitances, specifically the *C_BE_* of the input bipolar transistors of the comparator, from one input to the other; see Figure 6. Crosstalk at individual nodes of the resistor network degrades the integral nonlinearity, or the INL parameter, of the entire AD converter.

This problem is described in more detail in the literature [34]. Compared to the original 5-bit version of the ADC, the topology of the comparator has been replaced [16,33]. The comparator used in the 4-bit test cell consists of two differential amplifiers: a bipolar amplifier and a CMOS connected in series. An emitter follower is used as the input to maintain a high input resistance in the comparator and to shift the input voltage by the value of *V_BE_*. To achieve high gain, an active load was used. Figure 6 shows the circuit of a bipolar differential amplifier with an active load.

Based on the analysis of the differential amplifier, the voltage at one of the outputs is defined as $V_{0}=VCC-I_{C1}R_{C}$. In order to achieve a large voltage gain, $I_{C1}R_{C}$ must be as large as possible; this requires large supply voltages and large resistor values, which achieve large dimensions in the chip design. Replacing passive elements with active ones will reduce the number of passive elements on the chip, and the load size is then adapted to changes in the circuit. Transistors $M_{1}$ and $M_{2}$ form a current mirror controlled by transistor $M_{1}$. As a result, the reference current $I_{C1}$ is the same as $I_{C2}$. The output resistance of the amplifier depends on this current. The output voltage can be expressed as follows:(1)$$V_{o}=-\frac{1}{2}g_{m}(r_{o2}\left|\right|r_{o4})V_{id}$$
where $g_{m}$ is the current gain of transistors $Q_{2}$ and $r_{o2}\left|\right|r_{o4}$ are the output resistances of transistors $Q_{2}$ and $M_{2}$ [35,36]. In 0.35 µm SiGe BiCMOS technology, the PNP transistor structure is not supported, so PMOS transistors are used. A CMOS differential stage is used as the second stage because it better tolerates the operating point offset. If a bipolar differential stage is also used in this case, it would be necessary to readjust the operating point using emitter trackers, which would increase the power consumption. The output of the comparator consists of two CMOS inverters that amplify and shape the output signal. To adjust the operating point, current sources with NMOS transistors are implemented. These modifications over the original comparator circuitry increase the input range of the comparator to 1.6 V_pp_ and the gain of the comparator reaches 62 dB with a frequency bandwidth of 100 MHz (cutoff frequency −10 dB). The power consumption of one comparator reaches 5.75 mW at −3.3 V supply.

All synchronization and digital circuits are based on CMOS structures, which reduce the power consumption of the entire converter structure. The CMOS Master–Slave D-Flip-Flop (MSFF) circuits are used for synchronization [37,38]. The MSFF circuits synchronize the output of the comparators to the falling edge of the clock signal. Synchronization to the falling edge of the clock signal was deliberately chosen because this MSFF circuit topology achieves higher slew rates and higher speeds for the same output load. This is an acceleration of 120% for the rising edge and almost 200% for the falling edge; see Figure 7. This performance enhancement is due to the MSFF topology with synchronization to the falling edge, which features two CMOS inverters connected in series at the output, which increases the gain and the slew rate, and better forms the signal for the other circuits. In the case of rising edge synchronization, the output signal is taken from the first inverter [39].

To achieve the highest possible sampling rate, the clock distribution is divided into four blocks. The individual repeaters need to be placed as close as possible to the aforementioned D-flip-flop circuits. The output of the converter is represented by a thermometric code. Two CMOS inverters connected in series are used as output circuits. The output circuits directly decouple the flip-flop circuits and also provide sufficient excitation of the other circuits connected to the AD converter. The output is compatible with the −3.3 V LVCMOS standard [29]. The AD converter is implemented with additional circuit structures on a 2000 × 2000 μm chip. The core of the AD converter occupies an area of 1000 × 350 μm and the entire area of the structure, including the bonding contacts, is 2000 × 700 μm. The converter chip is contacted directly on the test PCB; see Figure 8a.

Compared to the previous version of the initially designed and implemented 5-bit AD converter, stable flipping of all comparators is achieved, and no errors are caused by the supply voltage drops in the internal structure of the chip. The setting of the comparators to an input voltage range of 1.6 V_pp_ is also confirmed, where the ADC has no problem processing the 1.6 V_pp_ input signal amplitude. A positive result is also observed in the maximum sampling frequency. By proper placement and reworking of the clock circuit layout, the maximum stable sampling rate increases to 1.1 Gs/s. The current consumption reaches 35 mA at 1.1 Gs/s with an effective number of bits (ENOB), i.e., 3.8 bits. The signal-to-noise and distortion (SINAD) ratio in this case is 24.83 dB. The values of INL and DNL nonlinearities are calculated by the historiographic method, as shown in Figure 8b, [21,40]. Detailed information on the 4-bit ADC structure is listed in [41]. This test structure serves as the basis for a new design of a 7-bit parallel AD converter implemented in 0.35 µm SiGe BiCMOS technology, intended for our UWB applications. An internal voltage reference is also implemented in the converter. This voltage reference is not connected to the AD converter directly on-chip but has a derived output for testing purposes.

## 4. Design of the 7-Bit AD Converter Structure

The topologies of parallel flash ADCs are quite familiar [34,42,43,44], but in specific cases, the concept can be adapted for application-specific requirements. The design of the 7-bit ADC resulted from the requirements summarized at the beginning of the chapter and from experience and measurements of the 4-bit test structure. A substantial part of the concept and the associated circuitry was taken only from the 4-bit test structure. From the dimensions of the test structure, the size and number of bits that could be squeezed into the 4 mm^2^ area (the desired user area based on the trade-off between performance and cost for the desired application) were tentatively determined. For the particular wiring with the UWB sensor system under consideration, it was necessary to adapt the I/O parts to be compatible with the analog circuitry of the system. Therefore, additional circuit solutions and structures were added. A simplified block diagram of the 7-bit ADC is shown in Figure 9. An explanation and definition of the individual inputs and outputs are presented in Table 1.

The AD converter consists of a so-called core, where the individual comparators and synchronizing D flip-flop circuits are located. The resistor network has both main nodes *VR_H* and *VR_L* routed out, through which, the upper and lower reference voltages can be controlled within the input range of the comparator. For output data compatibility with existing standards and FPGA connectivity options, the converter includes an encoder to convert the thermometry code to Gray code. This encoder consists of a 1 of N encoder and an encoder to the Gray code operating on the principle of ROM [30,34]. With an increase in structure size, the clock signal distribution network with differential inputs is also extended. The converter itself only has a non-differential input *RF_IN* for direct control and measurements, but also a standalone differential amplifier has been added to the AD converter chip. The output of the amplifier can be connected via the PCB trace to the input of the converter. This solution can provide differential excitation of the AD converter if needed.

### 4.1. Input Circuits, Development and Enhancements

The input block topology of the 7-bit ADC is almost identical to the previous test’s 4-bit ADC. The block diagram of the input part is shown in Figure 10. It contains basic blocks such as comparator and synchronization Master–Slave D-flip-flop (MSFF) circuit. The number of input circuits is 2*^N^* − 1, which represents 127 comparators and MSFF circuits. Because of better signal integrity, the 1 of N encoder has been added to the input block, from which the output goes directly to the ROM encoder.

The most important structure in a flash ADC is this input comparator. Schematic of the comparator used in the 7-bit ADC structure is shown in Figure 11. The parameters of the AD converter depend mainly on the parameters of the input comparators. Several topologies of comparators are based on bipolar or CMOS structures [45,46], but almost all of them work on the principle of the differential amplifier. Despite the very satisfactory results of the 4-bit ADC test structure, the input comparator was modified due to insufficient bandwidth and phase distortion. It should be noted that the decision level at a 1.5 V reference voltage and 4-bit ADC resolution for the least significant bit (LSB) flipping is as follows [47]:(2)$$V_{LSB}=\frac{V_{max}}{2^{N}-1}=\frac{1.5\;V}{2^{4}-1}=\frac{1.5\;V}{15}=100\;\mathrm{mV}$$
where *N* is the number of bits of the AD converter. As can be seen from the above result, the decision level reached values of 100 mV in the case of a 4-bit AD converter, which did not make it difficult to process tens of MHz where the comparator gain provided sufficient output switching. Based on Equation (Equation 2), the LSB decision level for a 7-bit converter and a 1.5 V reference voltage is around 11.7 mV. If it were necessary to reduce the reference range to 1 V, the LSB decision level would be around 7.8 mV.

However, this would be insufficient to drive the CMOS output buffers at a comparator gain of 35 dB (at 250 MHz). Moreover, the phase shift of the comparator in the 4-bit ADC at 100 MHz reaches 100°, Figure 12b. Based on experience with Cherry-Hooper amplifier designs, where these amplifiers achieve a 0.35 µm SiGe BiCMOS technology bandwidth over 10 GHz [48], the original comparator is replaced with a comparator based on the Cherry-Hooper structure [49].

The comparator includes an input buffer composed of emitter followers that separate the impedance of the input, thereby increasing the input impedance and simultaneously shifting the input signal by the value of *V_BE_* to adjust the operating point of the comparator. The core of the comparator consists of a Cherry-Hooper differential stage with active feedback [50,51]. The gain of the comparator’s differential stage was set at 42 dB as a compromise to maintain consistent switching characteristics over a bandwidth of at least 400 MHz. A comparison of the gain and bandwidth of the two versions of the comparators is shown in Figure 12a.

The output of the differential stage is again impedance-decoupled by an emitter follower, which also shifts the DC component of the amplified signal by the magnitude of the voltage *V_BE_*, thus, adjusting it for the proper switching of CMOS buffers. The output consists of two CMOS inverters that provide additional amplification and regeneration of the orthogonal signal. When testing the differential excitation of the comparator with a sinusoidal signal, it was found that the output duty cycle of the orthogonal signal does not reach 50%. Within the setting of the operating point of the differential stage and the offset of *V_BE_*, it was not possible to exactly achieve the center of the threshold voltage of *V_t_* CMOS inverters, causing a shift in the output signal. To achieve 50% of the output duty cycle, the selective width adjustment method of MOS transistors (STLS) from [38,52,53] was used. It is known that by adjusting different *W*/*L* ratios of MOS transistors, it is possible to change the threshold voltage of the *V_t_* CMOS inverter [52]. In cases where the transfer characteristic is not symmetrical, the logic threshold voltage can be approximated as follows [54]:(3)$$V_{t}=\frac{VDD-|V_{tp}|+V_{tp}\sqrt{K_{n}/K_{p}}}{1+\sqrt{P_{n}/K_{p}}}$$
where *VDD* is the supply voltage, *V_tp_* and *V_tn_* are the threshold voltages of PMOS and NMOS transistors, respectively. *k_n_* and *k_p_* are their transconductance coefficients given by [35,53]:(4)$$k_{n}=k_{n}^{\prime }{\left(W/L\right)}_{n}\;where:\;k_{n}^{\prime }=\mu_{n}C_{ox}$$
(5)$$k_{p}=k_{p}^{\prime }{\left(W/L\right)}_{p}\;where:\;k_{p}^{\prime }=\mu_{p}C_{ox}$$
where *k’_n_* and *k’_p_* are specified by the production process, *µ_n_* and *µ_p_* are the charge carrier mobilities and *C_ox_* is the oxide capacitance of the MOSFET transistor. Typical transconductance values *k’_n_* and *k’_p_* for SiGe BiCMOS technology are approximately *k’_n_* = 170 µA/V^2^ and *k’_p_* = 60 µA/V^2^. Based on these considerations, transistors *M*_1_ and *M*_4_ were set to *W* = 15 µm and the transistors *M*_2_ and *M*_3_ to *W* = 5 µm, achieving a 50% duty cycle at the output under symmetrical differential excitation; see Figure 13b.

Reducing the comparator gain increases the minimum LSB decision level to 2 mV_pp_ and the maximum hysteresis to 3 mV; see Figure 13a. The hysteresis is two times worse than the original comparator, but it is more stable over the entire input range of the comparator. The adjustment of CMOS output transistors also increases the comparator slew rate. A summary and comparison of the main parameters of the comparators for the 4-bit and 7-bit ADC are presented in Table 2.

The same Master–Slave D-Flip-Flop (MSFF) circuits are used to synchronize the output of the comparators. To these circuits, the 1 of N encoder was connected, as shown in Figure 14. To enhance the speed of the output CMOS inverters, the method of selectively adjusting the width of the MOS transistors was used, similar to the approach taken with the MSFF circuit and the 1 of N encoder. It should be noted that in the case of 0.35 µm SiGe BiCMOS technology, the charge carrier mobility *µ_p_* of the PMOS transistor is 0.35 times the mobility *µ_n_* of the NMOS transistor. If we know the current flowing through the MOSFET transistor given by equation [35], i.e.,
(6)$$I_{d}=\frac{k^{\prime }}{2}\frac{W}{L}{\left(V_{gs}-V_{t}\right)}^{2}$$
then in order to obtain the same current for both NMOS and PMOS transistors, based on Equations (Equation 4)–(Equation 6), it follows that
(7)$$k_{p}=k_{n}\;i.e.\;k_{p}^{\prime }{\left(W/L\right)}_{p}=k_{n}^{\prime }{\left(W/L\right)}_{n}$$
which indicates that the width of the PMOS transistor to achieve the same current as the NMOS transistor is increased by a multiple of the ratio of the transconductance values *k’*.

Increasing the current of PMOS transistors will increase the gain of the output inverters and accelerate the leading edge of the pulse. The negative effect of these modifications is an increase in the power consumption of the CMOS inverters. Therefore, only the transistors of the output inverters of the D-flip-flop circuit and the output of 1 of the N encoder were modified. In our design, the transistors *M*_2_ and *M*_4_ were set to a width of *W* = 5 µm, transistor *M*_1_ was set to a width of *W* = 10 µm, and transistor *M*_3_ was set to a width of *W* = 15 µm. All other transistors in Figure 14 have a width of *W* = 5 µm.

Figure 15 shows the on-chip layout and comparison of the input parts of the 4-bit and 7-bit ADCs. It is also possible to see the miniaturization of the entire structure, along with the addition of a 1 of N encoder.

More blocking capacitances have been added and the power supply branches have been modified for a more stable power supply to individual structures. The power supply modification helped to increase the scale of integration.

### 4.2. Clock Signal Distribution

The entire 7-bit ADC consists of 127 input parts that need to be properly synchronized. A schematic of the synchronization and clock signal distribution is shown in Figure 16. The converter is divided into eight four-bit parts, which are concatenated into the resulting 5-bit structures.

The clock signal is distributed from the input’s differential stage to four additional local differential amplifiers, which distribute the signal for the individual 5-bit structures. The schematic of the input’s differential amplifier is shown in Figure 17. The input’s differential amplifier amplifies the input clock signal and provides the functionality of the AD converter when both differential and non-differential clock signals are connected. The amplifier consists of input emitter followers that match the operating point for the differential stage while providing the ability to connect a clock with a zero DC component without the need to decouple the DC component. The input impedance is set using 50 Ω resistors.

The core of the input amplifier consists of a differential stage based on a Cherry-Hooper structure with emitter degeneration. This structure was selected for its wide bandwidth, necessary to drive up to four output stages configured as emitter followers. Connecting additional emitter followers capacitively loads the output of the differential stage, thus reducing the overall frequency bandwidth of the input amplifier. The input amplifier achieves a bandwidth of 11 GHz (−3 dB) at a gain of 18 dB. Sufficient bandwidth will ensure the same gain and orthogonal output signal even at frequencies around 1 GHz. The maximum output amplitude reaches 1.7 V_pp_.

The clock signal from the input amplifier is further regenerated and amplified by an additional differential stage for further local clock distribution. The schematic of the distribution amplifier is shown in Figure 18. The distribution amplifier consists of a simple differential stage that is modified by emitter degeneration and capacitive peaking to maintain a sufficient bandwidth. The DC operating point is set by the previous output. The output is provided by two pairs of emitter followers that shift the DC level of the clock signal for further processing by CMOS circuitry. The gain of the amplifier has been set to 10 dB and the frequency bandwidth reaches 10.5 GHz. The output signal amplitude reaches 1.85 V_pp_.

In each 4-bit section, the clock signal is distributed by a network of CMOS inverters. A detailed circuit diagram of the CMOS inverters is shown in Figure 19. Each section regenerates and amplifies a positive (*CLK*) or negative (*CLKN*) clock signal and distributes it to the D flip-flop circuits. These circuits require both positive and negative clock signals. Thus, we take advantage of the clock signal distribution using differential stages, where no additional CMOS inverter is required to reverse the polarity of the clock signal, if necessary. Another advantage is that a differential signal with a smaller amplitude than LVCMOS 3.3 V is passed through the entire chip. These differential signals almost cancel each other out and do not contribute as much to adverse interference and crosstalk on the chip. Also, in the context of the output CMOS inverters for the clock signal, it is necessary to selectively adjust the size of the transistor widths, to sufficiently excite multiple MSFF circuits.

### 4.3. ROM Encoder and Output Circuits

For further processing, the 128-bit thermometric code needs to be converted to a 7-bit code that can be connected to the FPGA module for further processing. Gray’s code was used as the output code due to better control of the output data [55]. In the Gray code, the nearby values (symbols) differ by only one bit, which can help in correcting the faulty bits [30]. The conversion of thermometric code to the Gray code can be conducted by various digital encoders (e.g., a ROM-based encoder [32], a Wallace tree encoder [56], a fat tree encoder [57], and the most widely used multiplexer-based encoder [58,59,60]); more information about the introduced encoders can be found in [30,31,61,62].

ROM encoders are among the simplest and fastest in terms of design and implementation, so a ROM encoder was chosen. A ROM is essentially a programmable logic device and can operate as a simple coding structure. It records a combination of input variables and generates an output for each combination. A ROM encoder acts as a hardware-based truth table for logic inputs and outputs. For use in an AD converter with a thermometric output, the thermometric code needs to be modified using a 1 of N encoder. The 1 of N encoder consists of NAND gates with one inverted input. A detailed schematic is shown in Figure 14. Wiring and connections to the ROM encoder are shown in Figure 20. Due to the difficulty and complexity of the 7-bit ROM encoder, a simple 3-bit structure is shown for clarity. The output buffer (consisting of three inverters) regenerates and amplifies the output data signal, and provides sufficient current capacity for data processing by other external circuits via the −3.3 V LVCMOS logic standard [29]. The maximum sink and source currents of the data outputs of the 7-bit ADC are 14 mA and 9 mA, respectively.

### 4.4. Auxiliary Amplifier

For the differential connection of the AD converter to the analog part of the UWB sensor system, an additional differential amplifier has been added to the AD converter chip, which has one output. This output is then fed back to the input of the converter through a trace connection on the PCB. This solution was chosen because of the ability to measure the AD converter directly, not through the amplifier. A schematic of the above amplifier is shown in Figure 21. The amplifier again consists of a differential stage based on a Cherry-Hooper structure and an output emitter follower. In all the circuits presented in this work, the current sources are created using NMOS transistors. The biasing of the current sources *IBIAS1* and *IBIAS2* are brought out to the bonding contacts to provide the possibility of setting the operating point according to the requirements of the analog part of the UWB sensor system and the AD converter input. The differential input is directly fed to the differential stage and has no internal DC bias to set the operating point.

Therefore, for proper functionality, it is important to set the required input bias voltage and input impedance externally, e.g., using resistors on the PCB. This external adjustment of the input operating point was built up because of the direct connection of the output of the analog part of the used UWB system, which adjusts the input operating point of the auxiliary amplifier. To provide the amplifier with an output range of 0 to −1.5 V (suitable for an AD converter), it was necessary to provide the amplifier with a positive supply voltage of *VCC*. The nominal value of the positive supply was set to 0.8 V, which corresponds approximately to the DC offset of *V_BE_* of the output’s emitter follower. A maximum gain of 13 dB can be set on the amplifier while the maximum frequency bandwidth is 19 GHz according to the simulation. The maximum output amplitude reaches 1.5 V_pp_.

### 4.5. Chip Layout and Arrangement of the 7-Bit ADC

The description and layout of the individual structures of the 7-bit ADC are shown in Figure 22. In Figure 22, four parallel 5-bit structures composed of comparators, D-Flip-Flop circuits, and encoder 1 of N can be seen. The main power supply of these 5-bit structures is routed from the bottom side and is separately distributed to the individual circuits. The individual input signals, such as the clock CLK signal and the analog input signal, are routed from the top side. This is also where their input amplifiers are located. The clock signal distribution is shown by the blue arrows. From the clock signal input amplifier, the other distribution amplifiers are excited.

These are evenly distributed to ensure the same clock signal for all circuits to the maximum extent possible. To eliminate crosstalk, emphasis has also been placed on the placement and crossover of analog, clock, and data signals, so that the signals are not routed in parallel, side by side. It can be seen that most of the signal traces cross at a 90° angle, which reduces crosstalk. In the middle is an ROM encoder whose outputs are fed to output buffers on the lower side of the chip. The free chip area has been filled with blocking capacitors. A total of 5644 transistors are implemented on the ADC chip. The entire chip is 2 × 2 mm in size and has been wire-bonded in a 6 × 6 mm QFN48 package. The unpackaged die of the AD converter is shown in Figure 23a and the wire-bonded AD converter in the package is shown in Figure 23b.

## 5. Development Board and Measurement

The converter chip contains many signal, power, and biasing pins, so such a converter could not be measured directly using the probes on the measuring station. For measurement purposes, a development board was designed. The block diagram is shown in Figure 24 and a photograph of the board during measurement is shown in Figure 25. The board contains the AD converter and the logic converters from a 3.3 V LVCMOS standard to a 3.3 V LVDS for the required signal integrity. The power supply and bias control are brought out on pin rails. The clock and input signals are applied to the board via SMP connectors. The output data signal is also brought out on the pin header.

In the design procedure of the development board, the focus was on the separation of the analog and digital ground plane and also on the impedance matching of analog and digital connections. For equal delay of the output digital data on individual bits, the lengths of all bus wires were adjusted to the same lengths (signal length tuning). For better high-frequency performance, the development board was fabricated on a double-sided RO4360G2 substrate with a dielectric constant of ε_r_ = 6.15 [63]. Due to the DC incompatibility, the output of the converter is DC-decoupled from the input of the LVDS converters.

As part of the measurement, two generators (Anritsu MG3700A and Keysight N5183B) were connected to the board for the input and clock signals. The two signals were fed single-ended and DC-decoupled by decoupling capacitors. The necessary power supply and bias voltages were set, e.g., for transistor biasing. The DC bias voltage of the converter was set via the output of the auxiliary differential amplifier to half of the reference voltage. The reference voltages from both the top and bottom sides, for the resistor network, were set using resistors and potentiometers. The output digital signal was acquired by a logic analyzer Tektronix TLA5201b, which was connected to the development board by a bus flat cable. Despite the effort to contribute to better digital signal integrity by using differential LVDS, the logic analyzer did not have a setting for impedance matching of the LVDS signal at the analyzer input. However, it was still able to analyze the individual logic levels. The output data were saved and processed using AD converter analysis scripts in Matlab and LabVIEW [64] environments.

In Figure 26, the results are shown of the measurement with an unexcited input sinusoidal signal with the frequency *F_IN_* = 1.156157 MHz and the amplitude *V_IN_* = 950 mV_pp_ at a sampling frequency of 200 Msps.

In such a measurement scenario, the converter achieves an effective number of bits, ENOB = 6.4 bits, a signal-to-noise and distortion ratio, SINAD = 36.54 dB, the strength ratio of the fundamental signal to the strongest spurious signal in the output, and the parameter spurious-free dynamic range, SFDR = 42 dBc. From the excited signal measurements and historiographic analysis, the integral and differential nonlinearities of the AD converter at different input sinusoidal signal frequencies and sampling frequencies were found. The results of the nonlinearity measurements are shown in Figure 27. The integral nonlinearity INL is a maximum of the 2-bit LSB and the differential nonlinearity DNL is a maximum of the 1.5-bit LSB. In these measurement scenarios, the converter was not equipped with a T&H input sampling circuit, which may have contributed to inferior results when compared to similar converters described in [65,66,67,68]. Calibration has not yet been performed on the presented converter. After calibration, the converter should exhibit significantly improved performance. The converter has also been tested for a maximum sampling rate, which reaches 1.4 Gs/s. The measured power consumption at 20 Ms/s is 1050 mW and at 1.4 Gs/s is 1290 mW, with a power supply of −3.3 V. A summary of the important parameters of the presented ADC and a comparison with other ADCs is presented in Table 3.

## 6. Conclusions

This article presents the development of UWB analog-to-digital converters for UWB M-sequence sensor systems and radars. In previous years, 5-bit and 4-bit parallel flash AD converters were designed as a part of UWB M-sequence systems research. Based on the knowledge gained by these initial designs, a 7-bit AD converter was designed, and fabricated specifically in low-cost 0.35 µm SiGe BiCMOS technology from AMS. As part of future developments, there are plans to create a radar comprising a transceiver and the proposed AD converter on one PCB, thereby verifying the overall compatibility of the analog part with the AD converter. Since the AD converter was designed using the same semiconductor technology as the developed UWB M-sequence radar transceiver, future work will be on its integration and compatibility with the analog part, specifically into a system-in-package (SiP) or directly into a system-on-a-chip (SoC). The proposed and fabricated application-specific 7-bit AD converter achieves the following parameters: ENOB = 6.4 bits, SINAD = 38 dB, SFDR = 42 dBc, INL = ±2-bit LSB, and DNL = ±1.5 LSB. The maximum sampling rate reaches 1.4 Gs/s, the power consumption at 20 Ms/s is 1050 mW, and at 1.4 Gs/s is 1290 mW, with a power supply of −3.3 V that is compatible with the UWB M-sequence radar transceiver.

## Figures and Tables

**Figure 1 sensors-24-02838-f001:**
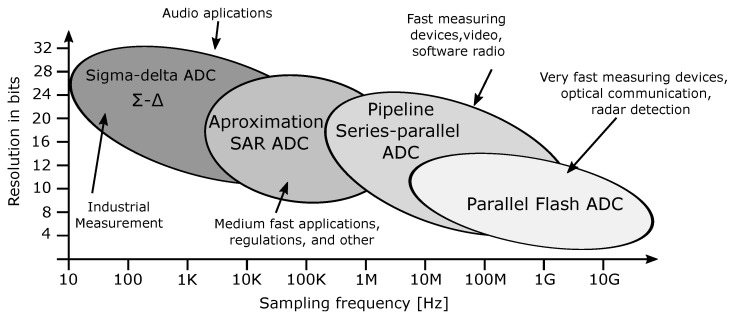
The AD converter architectures according to the sampling rate and resolution [11].

**Figure 2 sensors-24-02838-f002:**
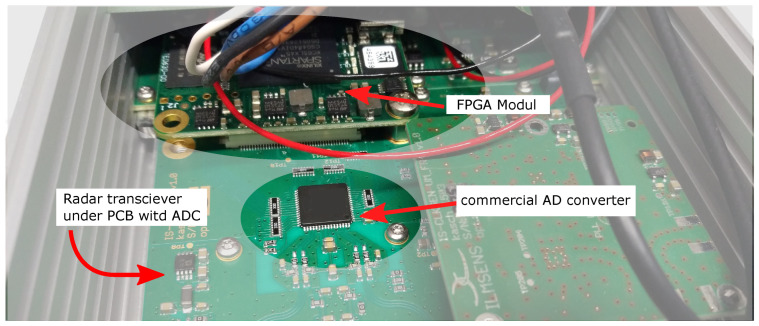
Exposed customized version of M-sequence UWB sensor system, m:explore [25].

**Figure 3 sensors-24-02838-f003:**
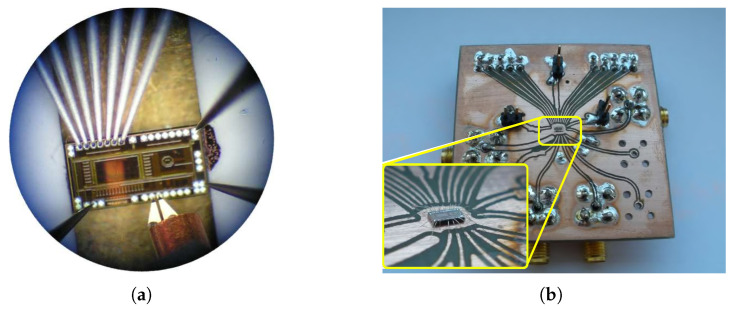
Designed a 5-bit parallel flash AD converter. (**a**) Measurement of the AD converter with the microprobe measuring station. (**b**) AD converter wire-bonded on PCB.

**Figure 4 sensors-24-02838-f004:**
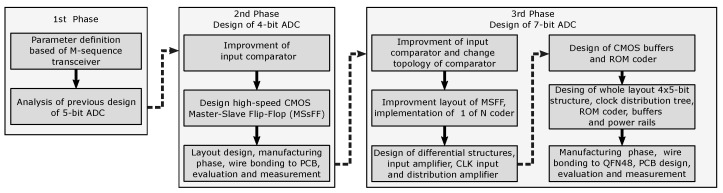
Flow chart showing the workflow of the ADC designs.

**Figure 5 sensors-24-02838-f005:**
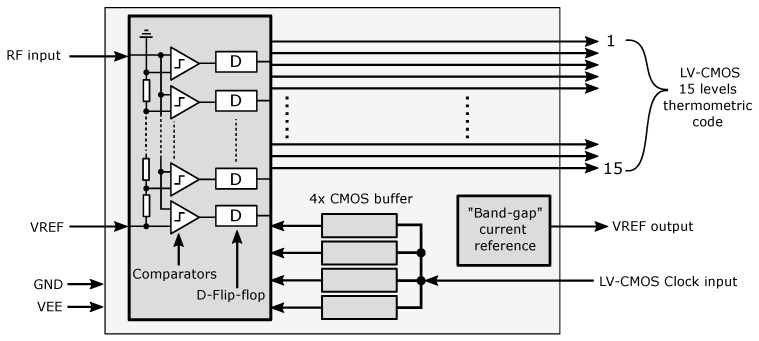
Concept of the 4-bit parallel AD converter structure.

**Figure 6 sensors-24-02838-f006:**
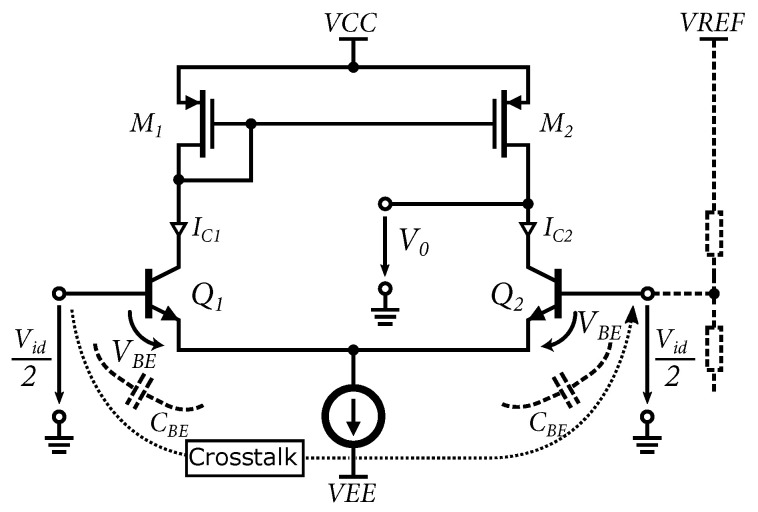
Circuit of the bipolar differential amplifier with active load and crosstalk indication.

**Figure 7 sensors-24-02838-f007:**
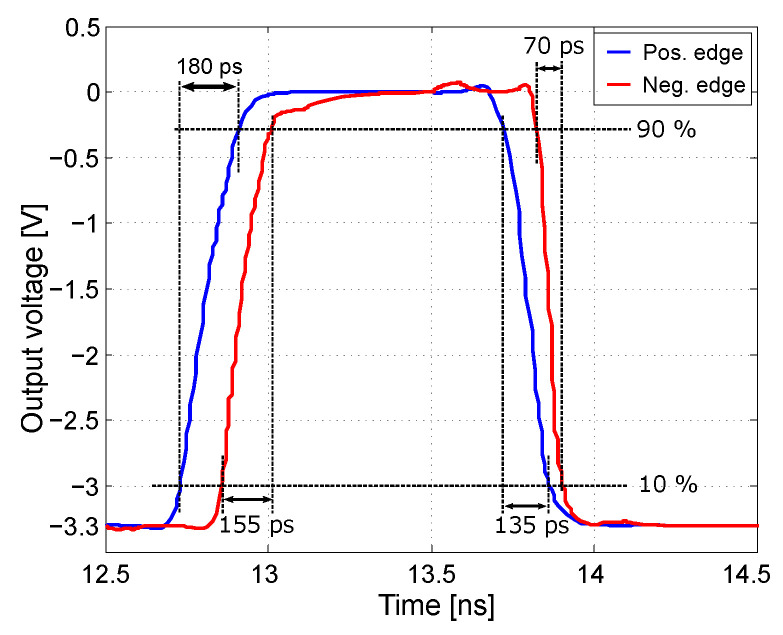
Simulation of MSFF circuits with synchronization to the positive and negative edges.

**Figure 8 sensors-24-02838-f008:**
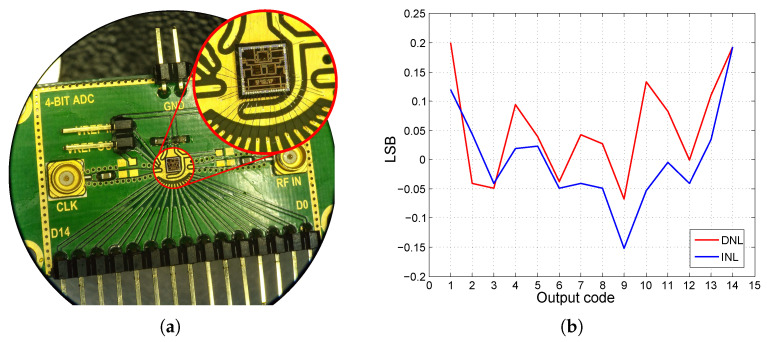
Designed 4-bit ADC cell. (**a**) View of the test PCB with the wire-bonded AD converter chip (**b**) Result of INL and DNL measurements, with 99.250 Ms/s sampling and 1 MHz input signal.

**Figure 9 sensors-24-02838-f009:**
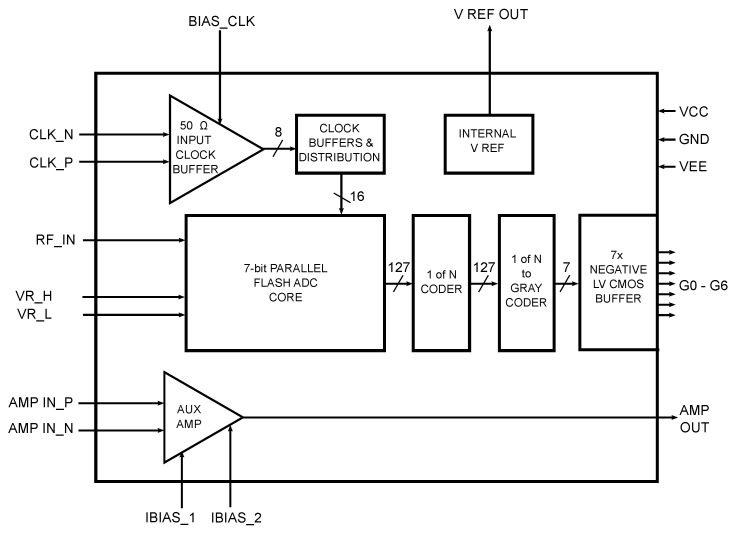
Simplified block diagram of 7-bit AD converter structure.

**Figure 10 sensors-24-02838-f010:**
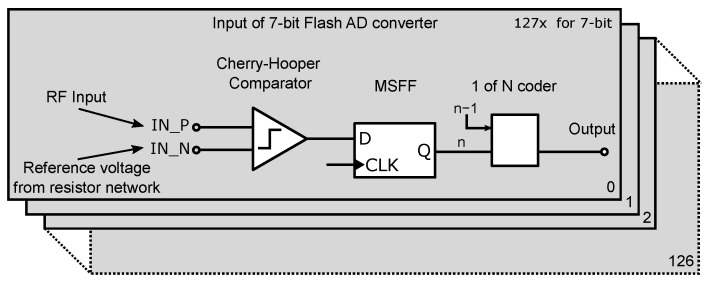
Input part of the 7-bit ADC structure.

**Figure 11 sensors-24-02838-f011:**
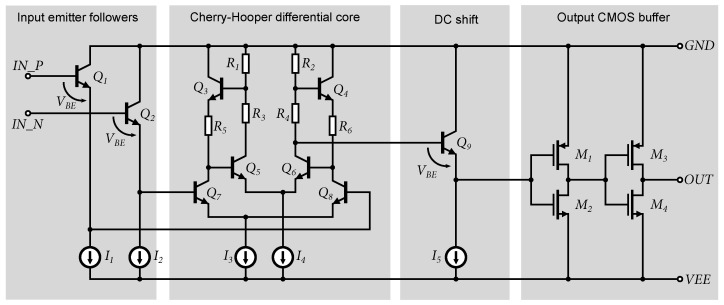
Input comparator based on Cherry-Hooper amplifier topology, with CMOS output buffer.

**Figure 12 sensors-24-02838-f012:**
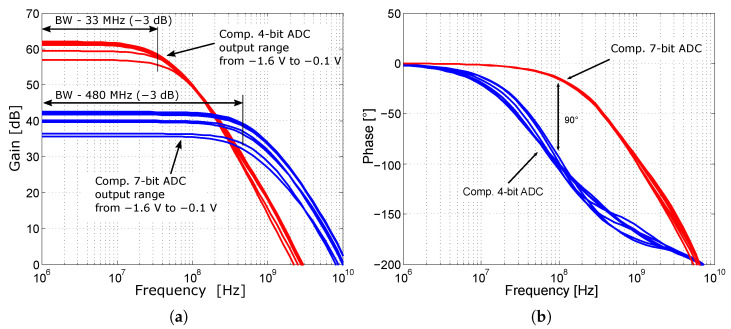
Results of the AC analysis of comparators at input range −0.1 V to −1.6 V. (**a**) Comparison of the gain and bandwidth of the analog part of the comparators. (**b**) Comparison of the output phase of the analog part of the comparators.

**Figure 13 sensors-24-02838-f013:**
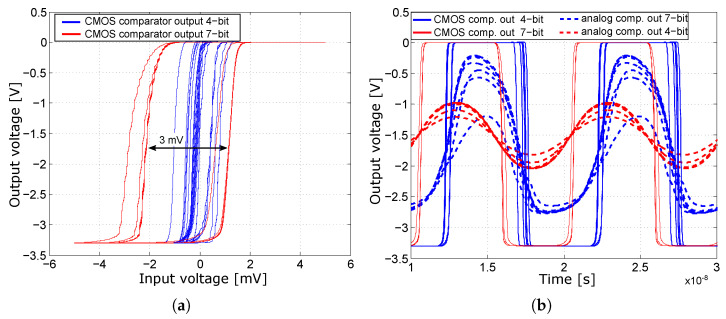
Characteristics simulated for an input signal with an amplitude of 5 mV_p_ and a frequency of 100 MHz, at an input range from −0.1 V to −1.6 V (**a**) Comparison of comparators—hysteresis. (**b**) Comparison of comparators—output waveforms.

**Figure 14 sensors-24-02838-f014:**
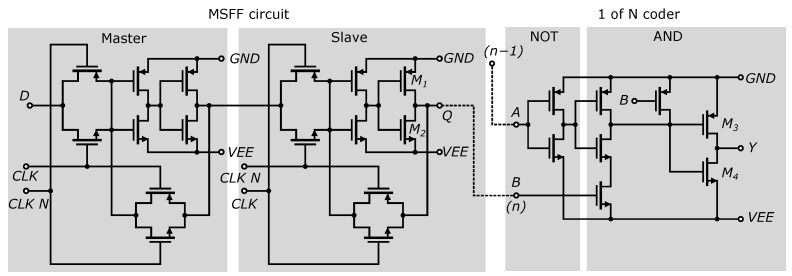
Used D flip-flop circuit and 1 of N encoder.

**Figure 15 sensors-24-02838-f015:**
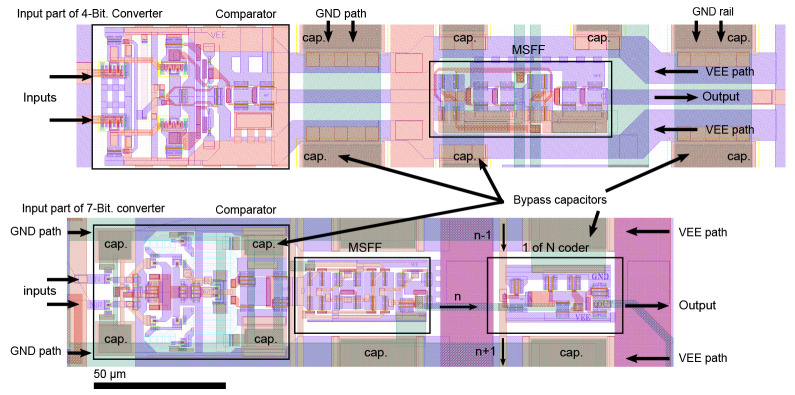
Layout comparison of the input sections of 4-bit and 7-bit AD converters.

**Figure 16 sensors-24-02838-f016:**
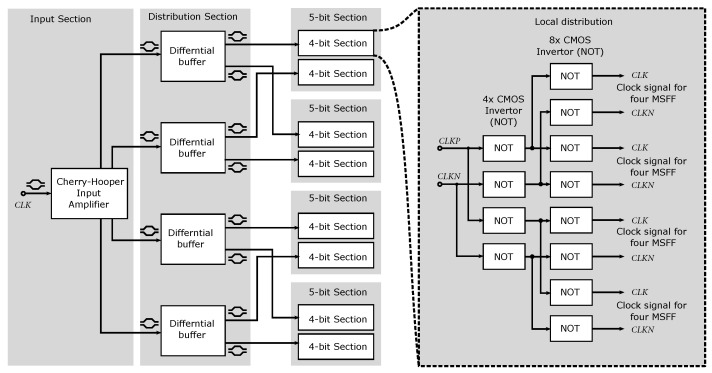
Block diagram of clock signal distribution.

**Figure 17 sensors-24-02838-f017:**
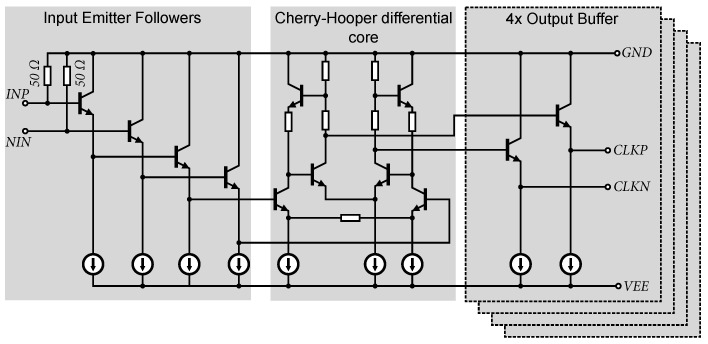
Circuit diagram of the clock signal input amplifier.

**Figure 18 sensors-24-02838-f018:**
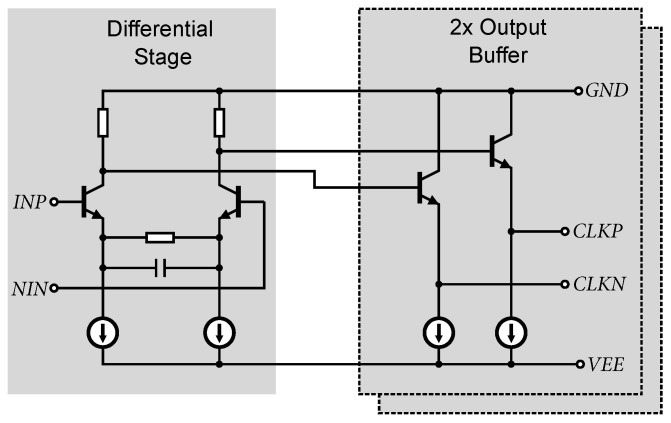
Circuit diagram of the distribution differential amplifier for clock signal regeneration.

**Figure 19 sensors-24-02838-f019:**
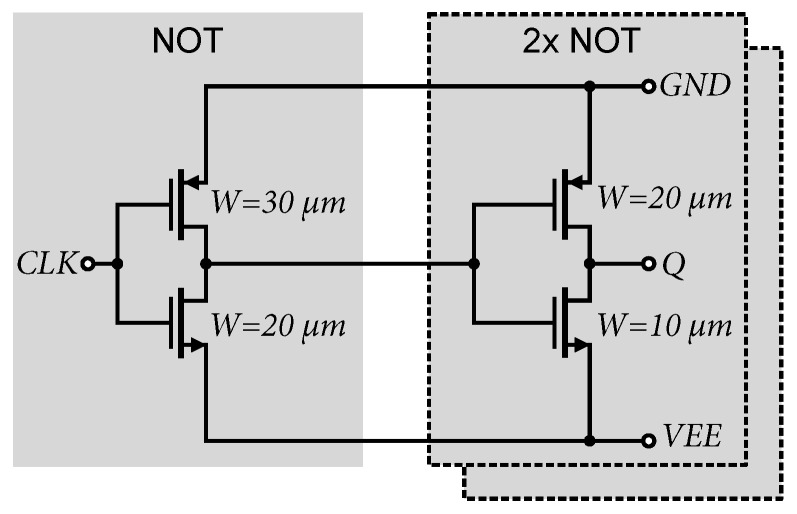
Connection of terminal inverters for clock signal distribution.

**Figure 20 sensors-24-02838-f020:**
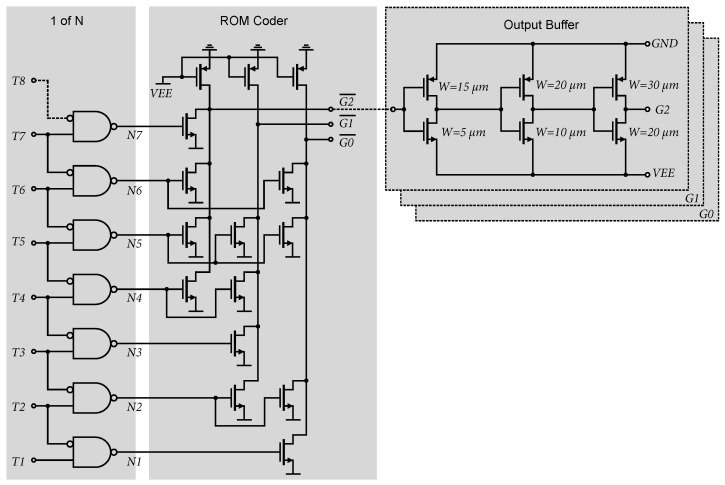
An illustrative example of a ROM encoder from the thermometric code to the Gray code for a 3-bit flash AD converter with connected outputs.

**Figure 21 sensors-24-02838-f021:**
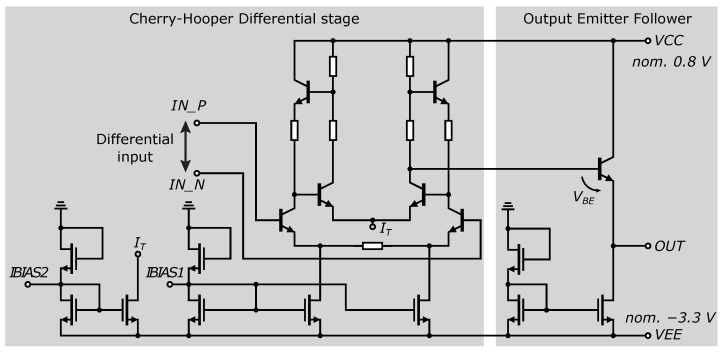
Schematic of the auxiliary differential amplifier.

**Figure 22 sensors-24-02838-f022:**
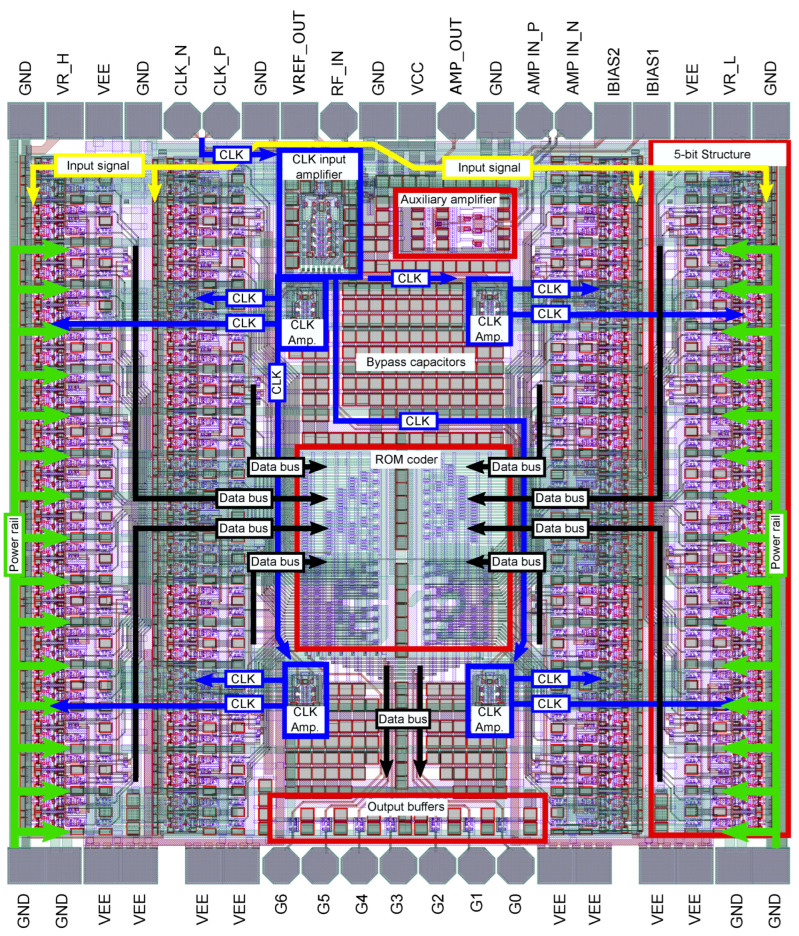
Layout of the 7-bit AD converter.

**Figure 23 sensors-24-02838-f023:**
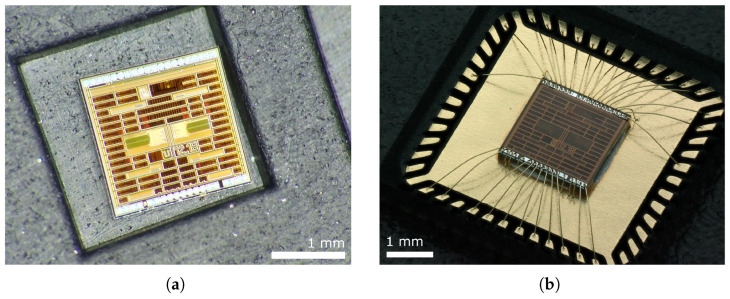
A photo of the manufactured 7-bit AD converter. (**a**) The manufactured die of a 7-bit AD converter. (**b**) Wire-bonded 7-bit AD converter in QFN48.

**Figure 24 sensors-24-02838-f024:**
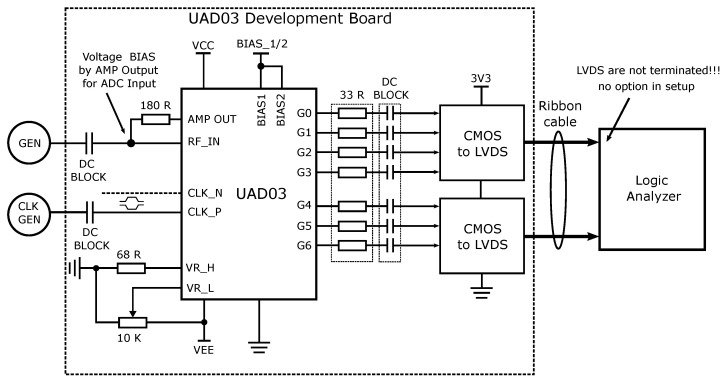
Block diagram of the 7-bit AD converter measurement on the development board.

**Figure 25 sensors-24-02838-f025:**
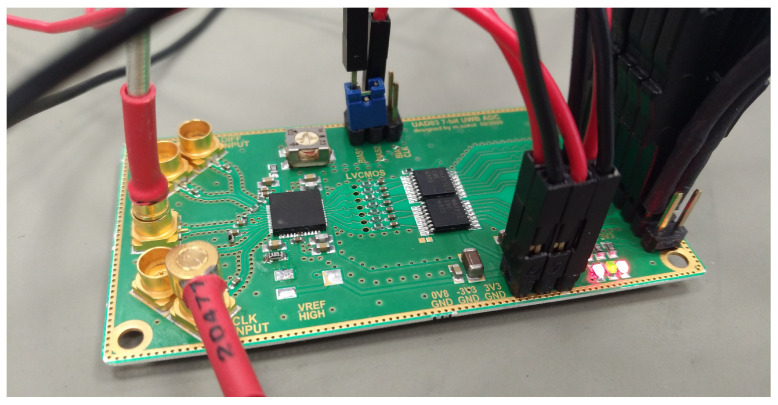
Evaluation PCB board of the 7-bit AD converter.

**Figure 26 sensors-24-02838-f026:**
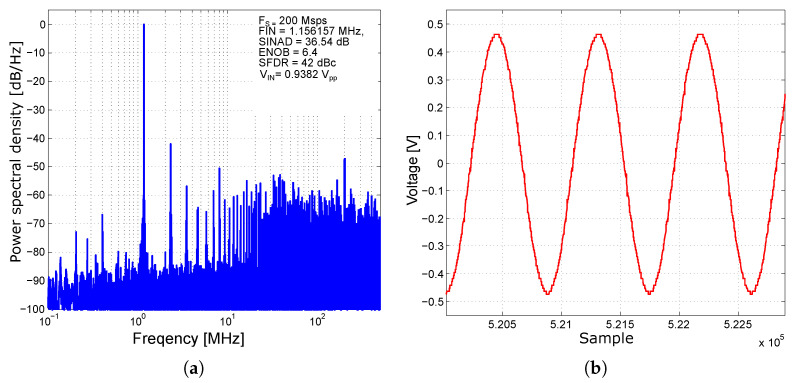
Measurement with an unexcited input sinusoidal signal at the frequency F_IN_ = 1.156157 MHz with amplitude V_IN_ = 950 mV_pp_ and at a sampling frequency of 200 Msps. (**a**) Power spectral density in a single-tone signal measurement. (**b**) Recorded signal in the time domain.

**Figure 27 sensors-24-02838-f027:**
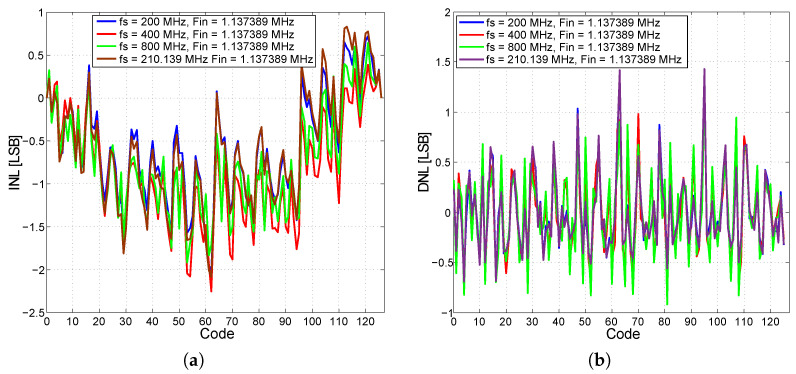
Results of INL and DNL nonlinearity measurements. (**a**) INL measurement result of the 7-bit AD converter. (**b**) DNL measurement result of the 7-bit AD converter.

**Table 1 sensors-24-02838-t001:** Name and definition of the individual inputs and outputs of the AD converter.

Pin Name	Type, Description
VCC	DC, positive supply voltage 0.8 V, for auxiliary amplifier
GND	DC, common ground
VEE	DC, common negative voltage −3.3 V
CLK_N, CLK_P	RF, differential clock input, AC or DC coupling
BIAS_CLK	DC, setting the operating point of the clock amplifier
RF_IN	RF, analog input of AD converter
VR_H	DC, high-side reference voltage
VR_L	DC, low-side reference voltage
G0–G6	RF, 7-bit parallel output in Gray code
AMP_IN_P, AMP_IN_N	RF, differential input of the auxiliary amplifier
AMP_OUT	RF, single-ended output of the differential amplifier
IBIAS1/IBIAS2	DC, setting the operating point of the differential amplifier
VREF_OUT	DC, current output of the internal band-gap reference

**Table 2 sensors-24-02838-t002:** Comparison of the main parameters of the comparators used for the 4-bit and 7-bit ADC.

Parameter	4-bit	7-bit
Supply voltage	−3.3 V	−3.3 V
Power consumption (RMS, 100 MHz)	4.2 mW	4.8 mW
Gain (DC)	62 dB	42 dB
Bandwidth (−3 dB)	33 MHz	480 MHz
Minimum resolution (100 MHz)	500 µV_p_	1 mV_p_
Slew rate	11.9 V/ns	14.4 V/ns
Input range (V)	1.6 V	1.6 V
Delay (100 MHz)	1.2 ns	623 ps
CMRR (100 MHz)	76 dB	73 dB
Hysteresis (100 MHz)	1.5 mV	3 mV

**Table 3 sensors-24-02838-t003:** A summary of the parameters of the presented AD converter and a comparison with other converters presented in [65,66,67,68].

Parameter	This Work	ADC07D1520 [65]	[66]	HMCAD1511 [67]	[68]
Semiconductor	0.35 µm BiCMOS	N/A	0.18 µm CMOS	N/A	0.25 µm SiGeC
Supply voltage	–3.3 V	2 V, 1.2 V	2.2 V	1.8 V, 3.3 V	2 V
Max. power consumption	1280 mW	1.9 W	711 mW	710 mW	2.6 W
Max. sampling frequency	1.4 Gsps	3 Gsps	4 Gsps	1 Gsps	3 Gsps
Bit resolution	7-bit	7-bit	4-bit	8-bit	6-bit
Input bandwidth	480 MHz	1 GHz	1.5 GHz	650 MHz	1.2 GHz
Input impedance, C_in_	750 Ω	100 Ω	N/A	11 pF	100 Ω
ENOB	6.5-bit	6.8-bit	3.89-bit	7.9-bit	4.5-bit
DNL	±1.5 LSB	±0.6 LSB	±0.15 LSB	±0.2 LSB	0.6 LSB
INL	±2 LSB	±0.9 LSB	±0.2 LSB	±0.5 LSB	0.6 LSB
Input voltage range	1.5V_pp_	940 mV_pp_	920 mV_pp_	2 V_pp_	500 mV_pp_
SFDR	42 dBc	45.5 dBc	36.5 dBc	49 dBc	50 dB
SINAD	38 dB	43 dB	N/A	45.7 dB	30 dB

## Data Availability

The data is availabe within the article.

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
