# Peer review of "Design of AD Converters in 0.35 µm SiGe BiCMOS Technology for Ultra-Wideband M-Sequence Radar Sensors"

_sensors, 2024, doi:10.3390/s24092838_

Round 1

Reviewer 1 Report

Comments and Suggestions for Authors In this work, the authors have designed a Flash ADC for UWB  application The  Comparators and ROM-based encoder used in the design are well known in the literature.  The novelty of the work lies in the fact that the chip fabricated has been integrated with the application  Also, the overall system functionality has been verified .   The reported work is a very good engineering effort and useful to the Radar Sensor community,   While the authors compared their design with two other existing designs, a direct comparison perhaps is fraught with danger since the technologies used are different. Also, the design with 0.18um CMOS process appears to perform better than the one by the authors.  Comments on the Quality of English Language

The English language used may be enhanced significantly after discussing with a language expert. Certain sentences are not framed correctly and usage of articles at some places in the text may not be appropriate. 

Author Response

Thank you for your review, we added more AD converters to comparison Table. 3. Please see the attachment.

Reviewer 2 Report

Comments and Suggestions for Authors

This paper is based on the design of AD converters in 0.35 µm technology and below are my comments for improving the manuscript.

1) The comparison table must be extended with more that 2 references and also with the recently published papers.

2) The future direction can be added to the conclusion section.

3) The authors must clarify that how do the initial design parameters are achieved and selected.

Author Response

Thank you for your review.
Comment 1.:
The comparison table must be extended with more that 2 references and also with the recently published papers.

Response 1: Thank you very much for your comment, we added more references and we extended the comparison Table 3.

Comment 2.: The future direction can be added to the conclusion section.

Response 2: Thank you very much for your suggestions to improve discussion, we added a next steps of our development in conclusion:

As a future development, there is a plan to create a radar comprising a transceiver and the proposed AD converter on one PCB, thereby verifying the overall compatibility of the analog part with the AD converter. Since the AD converter was designed using the same semiconductor technology as the developed UWB M-sequence radar transceiver, future work will be on its integration and compatibility with the analog part specifically into a system-in-package (SiP) or directly into a system-on-a-chip (SoC). Rows (490-495)

Comment 3.: The authors must clarify that how do the initial design parameters are achieved and selected.

Response 3: Thank you very much for your valuable comment. as mentioned in the article, the initial requirements were set based on the needs of our UWB M-Sequence radar system. They result from the M-Sequence transceiver, which was created firstly and based on that we created an AD-converter, which was then implemented on the SoC.

Reviewer 3 Report

Comments and Suggestions for Authors In this article, UWB analog to digital converters for UWB M-sequence sensor systems and radars are presented. This is a good effort from the authors. The reviewer's comments are: 

1- The abstract is written in an ambiguous style with poor English. It is recommended to be clear with your contribution in the Abstract.
2- The title of the paper needs to be revised. Define the words AD, UWB, etc. in the Abstract and Introduction.
3- In Table 1, how did the authors define the delay as small, medium, or large? What is the reference? Similarly, resolution, size, and sampling frequency are in this table. 

4- Please provide the ref for Fig. 2 5- Section 3.1 is too large, cut it down and provide mathematical modeling if possible.  6- It is recommended to add a flow chart showing the workflow. This is complex to follow with so many sub-parts.  7- What is the benefit of using negative voltage (-3.3V)?  8- Follow the MDPI template for Tables.  9- Most of the cited papers/references are old. It is recommended to add/replace references with the latest papers from the last 3 years at most.  Comments on the Quality of English Language

The English of the Abstract needs to improve. 

Author Response

Thank you for your review.

Comment 1- The abstract is written in an ambiguous style with poor English. It is recommended to be clear with your contribution in the Abstract.

Response 2: Thank you for your comment. We revised the abstract.

The article presents the analysis, design, and low-cost implementation of application-specific AD converters for M-sequence-based UWB applications to minimize and integrate the whole UWB sensor system. Therefore main goal is to integrate the AD converter’s own design together with the UWB analog part into the System in Package (SiP) or directly into the System on Chip (SoC), which cannot be implemented with commercial AD converters, or which would be disproportionately expensive. Based on the current and used UWB sensor system requirements, a parallel converter structure to achieve the maximum possible bandwidth in the proposed semiconductor technology was designed and presented in this article. A 5-bit and 4-bit parallel flash AD converters were initially designed as part of the research and design of UWB M-sequence radar systems for specific applications, and are briefly introduced in this article. The requirements for the newly proposed specific UWB M-Sequence systems were established based on the knowledge gained from these initial designs. After thorough testing and evaluation of the concept of the early proposed AD converters for these specific UWB M-sequence systems, the design of a new AD converter has been initiated. After confirming sufficient characteristics based on the requirements of UWB M-sequence systems for specific applications, a 7-bit AD converter in low-cost 0.35~µm SiGe BiCMOS technology from AMS was designed, fabricated, and presented in this article. The proposed 7-bit AD converter achieves the following parameters: ENOB = 6.4 bits, SINAD = 38 dB, SFDR = 42 dBc,  INL = ±2-bits LSB, and DNL = ±1,5 LSB. The maximum sampling rate reaches 1.4 Gs/s, the power consumption at 20 Ms/s is 1050 mW, and at 1.4 Gs/s is 1290 mW, with a power supply of -3.3 V.

Comment 2- The title of the paper needs to be revised. Define the words AD, UWB, etc. in the Abstract and Introduction.

Response 2: Thank you for your valuable comment. We revised the paper with native speaker, and we also defined all abbreviations in text.

Comment 3- In Table 1, how did the authors define the delay as small, medium, or large? What is the reference? Similarly, resolution, size, and sampling frequency are in this table.

Response 3: Thank you very much for your comment, the table 1. was created based on the source [11], but after reconsideration we decided to remove that table.

[11].  Rapuano, S.; Daponte, P.; Balestrieri, E.; Vito, L.D.; Tilden, S.J.; Max, S.; Blair, J. ADC parameters and characteristics. IEEE Instrumentation Measurement Magazine 2005, 8, 44–54. https://doi.org/10.1109/MIM.2005.1578617.

Comment 4- Please provide the ref for Fig. 2

Response 4: Thank you very much for your comment. We added more information about UWB radar sensor, see figure description:

Figure 2. Exposed customized version of M-Sequence UWB sensor system: m:explore [26].

[26] : Galajda, P.; Galajdova, A.; Slovak, S.; Pecovsky, M.; Drutarovsky, M.; Sukop, M.; Samaneh, I.B. Robot vision ultra-wideband wireless sensor in non-cooperative industrial environments. International Journal of Advanced Robotic Systems 2018, 15, 1729881418795767.

Comment 5- - Section 3.1 is too large, cut it down and provide mathematical modeling if possible.

Response 5: Thank you very much for your comment, we tried to reduce the entire chapter 3 in order to preserve the readability of the article. We also added a flow chart of development to chapter 3 (Figure 4.).

Comment 6- It is recommended to add a flow chart showing the workflow. This is complex to follow with so many sub-parts.  

Response 6: Thank you very much for your comment, we added a new flow chart about development of AD converter see. Figure 4..

Comment 7- What is the benefit of using negative voltage (-3.3V)?

Response 7: Thank you very much for your suggestions to improve the discussion, firstly is compatibility with existing UWB sensor circuits, also a simple 50 Ohm input matching with emitter follower, so it is possible to connect directly 50 Ohm resistor to GND.

Comment 8- Follow the MDPI template for Tables. 

Response 8: Thank you very much for your suggestions to improve our article. We have included your feedback in article tables.

Comment 9- Most of the cited papers/references are old. It is recommended to add/replace references with the latest papers from the last 3 years at most.

Response 9: Thank you very much for your suggestions to improve our article. We added new citations from last 3 years.

Round 2

Reviewer 2 Report

Comments and Suggestions for Authors

The authors have provided the comments and I have no further comments.

Reviewer 3 Report

Comments and Suggestions for Authors

No more comments.